# Prevention of Neurological Sequelae in Preterm Infants

**DOI:** 10.3390/brainsci13050753

**Published:** 2023-05-02

**Authors:** Eduardo Gonzalez-Moreira, Thalía Harmony, Manuel Hinojosa-Rodríguez, Cristina Carrillo-Prado, María Elena Juárez-Colín, Claudia Calipso Gutiérrez-Hernández, María Elizabeth Mónica Carlier, Lourdes Cubero-Rego, Susana A. Castro-Chavira, Thalía Fernández

**Affiliations:** 1Center for Biomedical Imaging and Neuromodulation, Nathan Kline Institute for Psychiatric Research, Orangeburg, NY 10962, USA; 2Unidad de Investigación en Neurodesarrollo, Departamento de Neurobiología Conductual y Cognitiva, Instituto de Neurobiología, Universidad Nacional Autónoma de México, Campus Juriquilla, Juriquilla, Querétaro 76230, Mexico; 3Laboratorio de Psicofisiología, Departamento de Neurobiología Conductual y Cognitiva, Instituto de Neurobiología, Universidad Nacional Autónoma de México, Querétaro 76230, Mexico

**Keywords:** preterm infants, MRI, Bayley’s II scores, Katona’s treatment, neurohabilitation, corpus callosum, lateral brain ventricles, prenatal risk, perinatal risk, neurodevelopment prediction

## Abstract

Background: Preterm birth is one of the world’s critical health problems, with an incidence of 5% to 18% of living newborns according to various countries. White matter injuries due to preoligodendrocytes deficits cause hypomyelination in children born preterm. Preterm infants also have multiple neurodevelopmental sequelae due to prenatal and perinatal risk factors for brain damage. The purpose of this work was to explore the effects of the brain risk factors and MRI volumes and abnormalities on the posterior motor and cognitive development at 3 years of age. Methods: A total of 166 preterm infants were examined before 4 months and clinical and MRI evaluations were performed. MRI showed abnormal findings in 89% of the infants. Parents of all infants were invited to receive the Katona neurohabilitation treatment. The parents of 128 infants accepted and received Katona’s neurohabilitation treatment. The remaining 38 infants did not receive treatment for a variety of reasons. At the three-year follow-up, Bayley’s II Mental Developmental Index (MDI) and the Psychomotor Developmental Index (PDI) were compared between treated and untreated subjects. Results: The treated children had higher values of both indices than the untreated. Linear regression showed that the antecedents of placenta disorders and sepsis as well as volumes of the corpus callosum and of the left lateral ventricle significantly predicted both MDI and PDI, while Apgar < 7 and volume of the right lateral ventricle predicted the PDI. Conclusions: (1) The results indicate that preterm infants who received Katona’s neurohabilitation procedure exhibited significantly better outcomes at 3 years of age compared to those who did not receive the treatment. (2) The presence of sepsis and the volumes of the corpus callosum and lateral ventricles at 3–4 months were significant predictors of the outcome at 3 years of age.

## 1. Introduction

According to Blencowe et al. [1], preterm birth is one of the world’s most critical health problems [2]. White matter injuries frequently accompany preterm birth, since preoligodendrocytes are highly vulnerable to hypoxia, ischemia, and inflammation, resulting in hypomyelination and a series of long-term events responsible for the neurologic and cognitive deficits observed in this population [3]. Extremely preterm infants are at a high risk of severe sequelae [4]. Infants born at less than 37 weeks of gestation are at an increased risk of abnormal neurodevelopmental outcomes, given that the central nervous system is susceptible to abnormal intra- and extra-uterine environments. Children born preterm are at risk of multiple neurodevelopmental sequelae involving dynamic and complex cognitive deficits [5].

The factors that influence neurodevelopment in infants born preterm are multiple, contributing to the complexity of follow-up research and variations in reported outcomes [6]. Prematurity and low birth weight (LBW) have been related to lower IQ in adulthood, demonstrating the enduring effect on cognition of these risk factors [7]. Negative outcomes of preterm birth can include cerebral palsy, motor impairments, intellectual disabilities, attention deficits, language impairments, autism, and psychiatric problems [8,9,10,11]. Preterm infants frequently present slow postnatal growth, which is related to the slow maturation of the cerebral cortex [12], and alterations of the cortical microstructure, which may explain their neurodevelopmental deficiencies [13].

In a previous study, Harmony [14] used Katona’s neurohabilitation therapy [15] to prevent neurologic and cognitive sequels in term and preterm infants with prenatal and perinatal risk factors for brain damage. The difference between neurohabilitation and neurorehabilitation is that, in neurohabilitation, the treatment should begin during the first months of life, before the lesion has been consolidated. By contrast, neurorehabilitation is usually initiated after sequelae have manifested.

Neurohabilitation is based on movements of the head that are intended to trigger a series of integrated complex movements, described by Katona as “elementary neuromotor patterns” that are unique to human beings and present from 25 weeks of gestational age (GA) to 6 months of life. The elementary neuromotor patterns are triggered by various head positions that activate the vestibular nuclei [15].

Our first objective, as reported in other studies, was to explore Bayley’s outcome at 3 years of age of a sample of 166 preterm infants, 128 of whom were treated with Katona’s therapy and 38 were untreated; all children had prenatal and perinatal risk factors for brain damage.

The main goal of this study was to explore the predictive value of risk factors and quantitative analyses of MRI obtained before 4 months of age for Bayley’s outcome at 3 years of age.

## 2. Materials and Methods

### 2.1. Ethics Statement

The Ethics Committee of the Neurobiology Institute of the National Autonomous University of Mexico (UNAM) approved this study (project 0.77H.), which also complies with the Ethical Principles for Medical Research Involving Human Subjects established by the Declaration of Helsinki. We obtained informed written consent to participate from the parents of all participants.

### 2.2. Participants

The sample was obtained from two main public hospitals in the state of Querétaro, the “Hospital de Especialidades del Niño y la Mujer” and the “Hospital del Instituto Mexicano del Seguro Social”, the hospitals where most of the children of Querétaro (México) are born and treated. After the infants were discharged from the hospital where they were born, their parents were invited to participate in a special study of the Unit for Neurodevelopmental Research (UNR) at the Universidad Nacional Autónoma de México (UNAM) in Querétaro. From a total of 934 infants that came to the UNR between 2004 and 2019, data for 166 preterm infants were examined. This sample of infants with prenatal and perinatal risk factors for brain damage had all the studies performed at similar ages including a corrected gestational age of two months or less. Exclusion criteria were age greater than 3 months at assessment, the presence of genetic factors associated with brain damage, cardiovascular pathology, gross brain malformations, and/or chromosomal aberrations, as well as infants with incomplete studies before 4 months.

Table 1 shows the distribution of the sample by weeks of gestational age (GA) and sex. These infants were assessed prior to treatment by the procedures described below and followed up to 3 years of age.

The parents of infants were invited to receive the Katona neurohabilitation treatment [16,17]. The parents of 128 infants accepted and received Katona’s neurohabilitation treatment. The remaining 38 infants did not receive treatment for a variety of reasons. At the 3-year follow-up, 128 infants who had undergone neurohabilitation were compared to 38 who had not.

### 2.3. Clinical Evaluations

Expert neuropediatricians clinically studied all infants following the recommendations of Amiel-Tison [18]. These specialists had access to the medical records from the hospital where the infants were born with data from the pregnancy and labor, as well as data from the newborn period of each infant. These data included prenatal and perinatal risk factors observed in the hospital. In the first neuropediatric examination of all infants, Katona’s maneuvers (see the description below) were examined, including startle reflexes and muscle tone abnormalities. The neuropediatricians also performed magnetic resonance images (MRI). These evaluations were repeated at different ages according to a fixed calendar up to 3 years of age.

### 2.4. Magnetic Resonance Image (MRI)

MRI before age 4 months was acquired during physiological sleep, thus, without sedation [19]. Brain MRI studies were acquired with a 3.0 T MRI scanner (General Electric Healthcare, Milwaukee, WI, USA). Structural images included axial T2-weighted FSE, coronal 3D T1-weighted SPGR, coronal 3D T2-weighted SE, and 3D-TOF SPGR. Brain MRI studies were evaluated and categorized according to neuroradiological patterns described in the literature (for an extensive review of perinatal brain damage patterns, see [19,20]), the temporality of the lesion according to [21], and the classifications and levels of severity according to white and gray matter abnormality scales [22,23].

Volumetry of the lateral ventricles and corpus callosum was obtained from the coronal 3D T1-weighted SPGR sequence. First, manual segmentation of the corpus callosum and lateral ventricles was performed according to well-known neuroanatomic boundaries described in detail in previous studies [19]. Afterward, estimates of the volumes and 3D reconstructions were determined. Volumes of the corpus callosum, as well as left and right lateral ventricles and subarachnoid space, indicating cortical atrophy, were quantified.

### 2.5. Treatment

Katona’s neurohabilitation method is diagnostic and therapeutic since the same maneuvers are used for both purposes. Treatment is initiated before three months of age to take advantage of higher brain plasticity before lesions have fully expressed. The procedure evaluates muscle tone (passive and active), posture and movement symmetry, attention, eye tracking, and auditory monitoring. If present, scissoring, strabismus, irritability, and axial hyperextension are considered abnormal. Katona described the elementary motor patterns. These patterns are chains of movements in which the neck, trunk, and extremities perform complex and continual movements in specific repetitive patterns.

At the beginning of the treatment, the therapist evaluates the infant using the various maneuvers described by Katona and obtains information required to program the exercises the parents will learn to perform at home. Each maneuver is intended to be repeated 4 to 5 times per session, and each session is generally composed of 6 to 8 maneuvers. This intensive treatment consists of three or four 45-min sessions per day. Treatment outcome presumably depends on the intensity and accuracy of the personalized training [14,15]. Parents are required to visit the therapist daily for supervision during the first 3 months. After that age, the therapist sets the frequency of subsequent evaluations. The therapist also organizes a daily cycle to integrate the sessions into the infant’s sleep-wake schedule and feeding and nursing periods. Every month, the infant is examined to determine the ages at which the infant has mastered various developmental milestones. Monthly evaluations are scheduled to update the maneuvers that parents should learn to perform at home. After the infant walks independently, treatment continues with a focus on fine movements, until about 24 to 36 months of age.

### 2.6. Bayley Scale of Infant and Toddler Development

This scale [24] assesses the neurodevelopment of young children, from 1 to 42 months of age. The Bayley’s II battery was administered to all 166 infants, including 128 treated and 38 untreated infants at 4 months and when the infants reached 3 years of age, yielding a Mental Developmental Index (MDI) and a Psychomotor Developmental Index (PDI).

### 2.7. Statistical Analysis

We used multivariate permutation comparisons at the age of 4 months to evaluate differences between treated and untreated infants, applying correction for multiple comparisons. These comparisons were carried out for risk factors and MRI variables. We also used permutation analysis to explore differences between treated and untreated children at the age of 3 years on MDI and PDI.

We were also interested in knowing if the Bayley scores obtained at the age of 3 years can be predicted by the risk factors and MRI characteristics. Therefore, we calculated linear regressions for the MDI and PDI at 3 years of age using all prenatal and perinatal risk factors and MRI features (specifically, volumes of corpus callosum, and lateral ventricles) as predictors. For this purpose, linear regressions considering MDI and PDI at 3 years age as dependent variables were performed. The data of this study had a multivariate nature, so correction for multiple comparisons was necessary to avoid the inflation of type I and type II errors.

## 3. Results

### 3.1. Risk Factors

All infants have prenatal and perinatal risk factors for brain damage. Within the prenatal risk factors, the most frequently observed were mother’s infections of the genitourinary tract (39%), toxemia (22%), and placenta abnormalities (14%) in the total sample. Previous abortions and diabetes were also observed.

Table 2 shows the prenatal risk factors for each gestational age and the treated and untreated subjects. There were no significant differences in the number of subjects for each risk factor between the treated and untreated groups. Maternal infections and toxemia during pregnancy were the most frequent prenatal factors.

Perinatal risk factors were newborn’s sepsis (65%), the need for oxygenation (66%), and hyperbilirubinemia (62%) that was treated with phototherapy with excellent results; metabolic disorders, seizures, respiratory distress, growth restriction in utero, preterm retinopathy, anemia, necrotizing enterocolitis, growth restriction in utero, respiratory distress, congenital heart anomalies, preterm retinopathy, anemia, necrotizing enterocolitis, and hypoxic-ischemic encephalopathy were also observed.

Table 3 display the perinatal risk factors for brain damage at each gestational age for the treated and untreated infants. For the risk factors in Table 3, no significant differences were observed between the number of treated and untreated infants for each risk factor.

### 3.2. MRI

Magnetic resonance images (MRI) are considered the gold standard for clinical diagnosis. For this reason, we included them in evaluating the patients to know if they have abnormal structural brain findings. Clinical interpretation of the MRIs showed that only 19 patients (11%) were considered normal for the total sample. The most frequent observation was the increased volume of the liquid spaces (lateral ventricles and subarachnoid space) and the reduced volume of the corpus callosum. More than 50% of the infants from the extremely preterm and very preterm showed volumes of the corpus callosum smaller than 1 mL^3^. Infants with diffuse white matter injuries, periventricular leukomalacia, and periventricular hemorrhages were in these groups. Periventricular hemorrhages were more frequently observed in the extremely preterm group than in the other groups. Hypoxic ischemic encephalopathy and brain infarcts were observed in the late preterm infants. Table 4 shows the mean and standard deviation of the volumes of the corpus callosum and the right and left lateral ventricles.

Figure 1, Figure 2 and Figure 3 show the MRI of three infants at the beginning of Katona’s treatment.

### 3.3. Bayley’s Scales Results

Bayley’s II scales [24] were administered at three years of age in all infants except for one moderate preterm. We used multiple permutation tests for the comparisons between infants treated with Katona’s procedure and those untreated. The result of the multiple comparison between those treated and untreated were significant at *p* = 0.001, with the group of children treated with greater values of the MDI and the PDI. Table 5 shows the results.

At three years of age, clinical evaluations were conducted on preterm infants across four groups. Among the extremely preterm group, 8 of 9 treated infants had typical cognitive and motor development, while only 1 out of 4 untreated infants showed adequate development. Similarly, 32 out of 42 treated infants and 11 out of 15 untreated infants in the very preterm group had typical cognitive and motor evaluations. In the moderate preterm group, the percentage of treated (33 out of 41) and untreated (11 out of 15) infants with typical cognitive evaluations was similar. However, untreated infants exhibited motor impairments such as coordination deficits and monoparesis. In the late preterm group, untreated infants showed more frequent deficits in both cognitive and motor areas than treated infants.

Of crucial relevance was the finding that receiving Katona treatment is essential in achieving better Bayley scores at 3 years of age.

For us, it was also important to know the predictive value for Bayley’s MDI and PDI of the risk factors and MRI volumes. The significant predictors are shown in Table 6 For both MDI and PDI, the significant predictors of risk factors were the placenta disorders and the presence of sepsis in the infant; as the risk factors increased, Bayley’s indices decreased. PDI significantly decreased when Apgar < 7 at 5 min and when oxygen was needed at birth. The significant MRI predictors for lower scores on both indices were a reduced corpus callosum and a dilated left lateral ventricle; a dilated right lateral ventricle significantly predicted a lower PDI.

## 4. Discussion

### 4.1. Sample Characteristics and Prenatal and Perinatal Risk Factors

Preterm infants present more risk factors for brain damage than at-term infants [25]. In our sample, all infants have pre and/or perinatal risk factors. Maternal infections (39%) were the most frequent prenatal risk factor found in this study, almost all from the genitourinary tract; this agreed with a recent study by the World Health Organization [26].

Other factors were toxemia during pregnancy (22%) and previous abortions (14%). Toxemia is a severe risk factor [27] associated with fetus undernutrition and growth retardation [28]. Preeclampsia accounts for almost 6–10% of gestational complications and has become increasingly prevalent in recent decades [29]. Previous abortions are generally related to preterm delivery [30,31].

The perinatal risk factors most frequently observed were sepsis (65% in the total sample) and the need for oxygen administration (42%). Sepsis remains a significant cause of morbidity and mortality among neonates, as observed in a publication reporting 75 years of study at Yale [32]. Significant risk factors for poor outcomes of neonatal sepsis were respiratory distress syndrome and meconium aspiration syndrome, as reported by [33]. Additionally, the perinatal risk factors we observed, Apgar < 7 at 5 min and seizures in 19% and 13% of infants, respectively, showed the severity of brain damage in our sample since both are indices of poor outcomes. Seizures indicate metabolic disturbances, hypoxic conditions, intracranial hemorrhage, severe infection, arterial ischemic stroke, and sinus venous thrombosis [34].

Another perinatal factor observed was growth restriction in-utero. Fortunately, it was infrequent since it has as consequences altered heart structure, pulmonary dysplasia, and, in the brain, decreased total volume of both grey and white matter and altered cerebral connectivity [35].

Another factor, respiratory distress, is one of the most common risk causes in newborns, including transient tachypnea of the newborn (TTN), respiratory distress syndrome (RDS), meconium aspiration syndrome, pneumonia, sepsis, pneumothorax, and delayed transition [36]. However, newborns’ most common congenital anomalies are cardiac [37]. Heart anomalies are present in 5 to 8 of every 1000 newborn babies. Many of these do not require any immediate treatment. For example, those anomalies that are smaller than the openings (ventricular septal defect) in the wall between the two ventricles are mostly closed later.

In our sample, respiratory distress, heart anomalies, and necrotizing enterocolitis (NEC) were observed. NEC was more frequent in the extreme and very preterm groups, as reported previously [38]. The reported incidence varies from one country to another, ranging from 5 to 15% [38]. Anemia and its significant impact on cerebral tissue oxygenation was also observed. Preterm anemia is associated with a pathological decline in tissue mixed with venous saturations and is a proposed mechanism for NEC [39].

MRIs were abnormal in 89% of the total sample, indicating that the risk factors observed impacted the brain structure. This result agrees with [14]. Reduction of the volume of the corpus callosum (CC) was observed in 50% of the sample. Abnormalities in the CC are related to impaired motor and delayed cognitive development [40,41]. Our results showing that CC volume was a predictor for both MDI and PDI support these previous observations.

Increases in the volumes of the liquid spaces were frequently observed; this factor has been associated with the outcome at 24 months [42] and is accompanied by compression of white matter and some degree of cerebral cortex atrophy. Volume of the left lateral ventricle was a predictor for both Bayley’s MDI and PDI, and the volume of the right lateral ventricle predicted PDI. These results may be explained considering that compression of the left white mater by corresponding ventricle directly affects the areas related with language and, therefore, the MDI. On the other hand, compression of the right white matter by an increased volume of the right lateral ventricle impacts gross and fine motor control and coordination and visual and spatial tasks.

Additionally, non-cystic PVL was observed in the preterm groups. Our findings agree with those previously reported [21,43,44].

### 4.2. Effect of Katona’s Neurohabilitation Treatment

Katona’s treatment is based on the activation of the vestibular receptors from different head positions. The outputs of the semicircular canals activate the vestibular nuclei that have ascending and descending connections to several brain stem nuclei, the reticular formation, the thalamus, the cerebellum, the basal ganglia, and the medulla. At the same time, these movements activate proprioceptors that send direct connections to the cerebral cortex and the cerebellum. This continuous feedback is believed to help the organization of motor control [14,15].

Previous reports have shown the beneficial effect of Katona’s neurohabilitation treatment during infancy in children with perinatal brain lesions followed up 6–8 years of age compared with children who voluntarily discontinued treatment after the initial evaluation before 2 months of age [19]. In another study [45], two groups of infants were formed; in one group, Katona’s procedure was used, and the other received “neurodevelopmental therapy” [46]. Both groups were matched by sex, gestational age, and lesion severity observed on MRI, and both procedures were initiated before three months of corrected age. Infants were followed up to five years of age or more, and, at this age, a blind evaluation of the two groups of children was performed by an expert neuropediatrician. Comparisons of outcomes showed a higher proportion of children with normal neurodevelopment were treated with Katona’s (9/11) than those treated with the “neurodevelopmental therapy” (4/11).

Harmony [14] reported that 78% of extreme preterm, 76% of very preterm, and 78% of late preterm had normal neurodevelopment. In the present manuscript, the comparison of the MDI and PDI indices at 3 years of age, between treated and untreated infants, showed significant greater values for the treated group, corroborating our previous findings.

### 4.3. Limitations of the Study

The principal study limitation was that we did not randomly assign infants to be treated versus untreated. However, according to the World Medical Association Declaration of Helsinki, the use of a placebo or no intervention of the condition is acceptable only when no proven intervention exists. Given prior evidence that Katona’s neurohabilitation treatment has been helpful in treating infants with brain damage to prevent sequelae [14,15], treatment was offered to all infants, and most parents accepted it. Fortunately, the groups of treated and untreated infants did not differ significantly on any of the prenatal or perinatal risk factors.

## 5. Conclusions

(1) The results indicate that preterm infants who received Katona’s neurohabilitation procedure exhibited significantly better outcomes at 3 years of age compared to those who did not receive the treatment. (2) The presence of sepsis and the volumes of the corpus callosum and lateral ventricles at 3–4 months were significant predictors of the outcome at 3 years of age.

## Figures and Tables

**Figure 1 brainsci-13-00753-f001:**
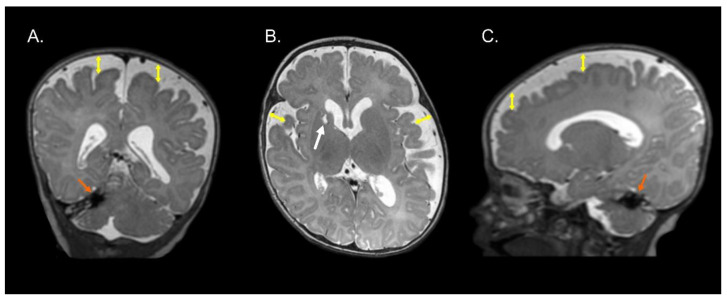
Coronal (**A**), axial (**B**), and sagittal (**C**) T2-weighted structural MRI of a preterm infant (born at 34 gestational weeks) at 2 months of corrected age with right cerebellar hemorrhage (orange arrows) and ischemic process with cystic degeneration in head of right caudate nucleus (white arrow). Note an enlargement of the lateral ventricles (left predominance) and increase in the subarachnoid space in the fronto-temporo-parietal region (double headed yellow arrows). Images in radiological convention.

**Figure 2 brainsci-13-00753-f002:**
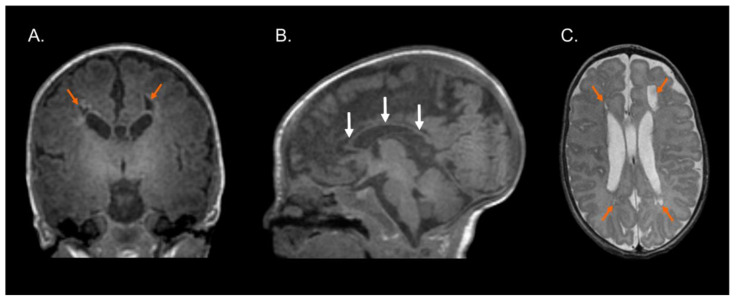
Structural MRI at term-equivalent age of preterm infant (born at 29 gestational weeks) with cystic-PVL. T1-weighted coronal (**A**) and sagittal (**B**) images and T2-weighted axial image (**C**) showing bilateral cystic white matter lesions (orange arrows), global thinning of the corpus callosum (white arrows), dilated lateral ventricles secondary to cerebral white matter loss, and myelination process delay. Images in radiological convention.

**Figure 3 brainsci-13-00753-f003:**
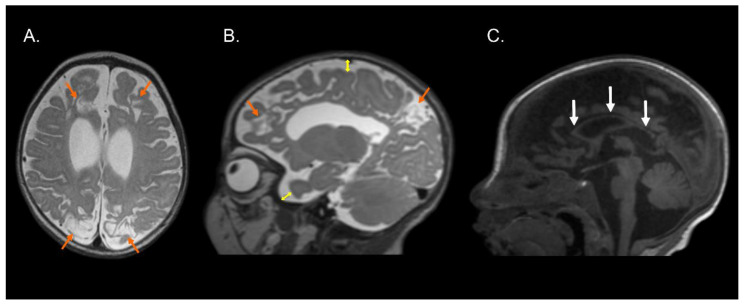
Structural MRI of preterm infant (born at 36 gestational weeks) with a diagnosis of perinatal hypoxic-ischemic encephalopathy. T2-weighted axial (**A**) and sagittal (**B**) images and T1-weighted sagittal image (**C**) exhibiting bilateral parasagittal encephalomalacia in fronto-parieto-occipital region (orange arrows), dilated lateral ventricles, augmented subarachnoid space in the fronto-temporo-parietal region (double direction yellow arrows), and hypoplasia of the corpus callosum (white arrows). Images compatible with parasagittal cerebral injury or “watershed injury”. Images in radiological convention.

**Table 1 brainsci-13-00753-t001:** Sample characteristics.

Group	N	Gestational Age (Weeks)	Male/Female
Extremely preterm	13	25–27	7/6
Very preterm	57	28–31	35/22
Moderate preterm	51	32–34	27/24
Late preterm	45	35–37	27/18
TOTAL	166	25–37	96/70

**Table 2 brainsci-13-00753-t002:** Prenatal Risk Factors.

Groups	Total Sample		Maternal Age >40 Years		Previous Abortions		Maternal Infections		Toxemia		Diabetes		Placenta Alterations	
	N	%	N	%	N	%	N	%	N	%	N	%	N	%
Extremely preterm	13	8	0	0	3	31	5	62	2	8	0	0	0	0
Treated	9	7	0	0	2	21	4	52	1	4	0	0	0	0
Non-treated	4	10	0	0	1	10	1	10	1	4	0	0	0	0
Very preterm	57	34	2	4	10	18	20	35	13	23	1	2	9	16
Treated	42	33	2	4	8	14	17	27	10	21	1	2	6	11
Non-treated	15	39	0	0	2	4	3	8	3	2	0	0	3	5
Moderately preterm	51	31	2	4	5	10	20	39	15	29	0	0	5	10
Treated	41	32	2	4	3	6	15	30	12	25	0	0	4	8
Non-treated	10	26	0	0	2	4	5	9	3	4	0	0	1	2
Late preterm	45	27	2	4	6	13	20	44	6	13	2	4	9	20
Treated	36	28	1	2	3	7	16	38	3	7	2	4	6	13
Non-treated	9	24	1	2	3	7	4	6	3	7	0	0	3	7
Total	166	100	6	4	24	14	65	39	36	22	3	2	23	14
Treated	128	77	5	4	16	13	52	31	26	14	3	2	16	10
Non-treated	38	23	1	1	8	5	13	8	10	8	0	0	7	4

**Table 3 brainsci-13-00753-t003:** Perinatal Risk Factors.

		Extremely Preterm			Very Preterm			Moderately Preterm			Late Preterm			TOTAL		
		N	T	NT	N	T	NT	N	T	NT	N	T	NT	N	T	NT
Total	N	13	9	4	57	42	15	51	41	10	45	36	9	166	128	38
	%	5			23			20			18			100		
Metabolic disorders	N	4	3	1	12	10	2	5	3	2	7	4	3	28	20	8
	%	31			21			10			16			17	16	21
Apgar < 7	N	6	5	1	7	4	3	8	6	2	10	6	4	31	21	10
	%	46			12			16			22			19	16	26
Oxygen	N	10	9	1	21	15	6	24	23	1	15	13	2	70	60	10
	%	77			37			47			33			42	47	26
Sepsis	N	12	8	4	45	31	14	31	27	4	20	17	3	108	83	25
	%	92			79			62			44			65	65	66
Seizures	N	2	2	0	7	4	3	6	4	2	6	3	3	21	13	8
	%	16			12			12			13			13	10	21
Neonatal jaundice	N	11	8	3	36	26	10	36	28	8	27	24	3	110	86	24
	%	85			63			71			60			66	67	63
Phototherapy	N	11	8	3	32	24	8	33	26	7	25	22	3	101	80	21
	%	85			56			65			56			61	63	55
Growth restriction in utero	N	0	0	0	5	3	2	3	3	0	4	4	0	12	10	2
	%	0			9			6			9			7		
Respiratory distress	N	3	3	0	10	8	2	7	6	1	3	2	1	23	19	4
	%	23			18			14			7			14		
Congenital heart anomalies	N	2	1	1	7	5	2	4	4	0	1	1	0	14	11	3
	%	16			12			8			2			8		
Preterm retinopathy	N	1	1	0	7	6	1	0	0	0	0	0	0	8	7	1
	%	8			12			0			0			5		
Anemia	N	0	0	0	7	7	0	3	3	0	4	4	0	14	14	0
	%	0			12			6			9			8		
Necrotizing enterocolitis	N	0	0	0	4	4	0	0	0	0	1	1	0	5	5	0
	%	0			7			0			2			3		
Encephalopathy hypoxic ischemic	N	0	0	0	2	2	0	3	2	1	3	3	0	8	7	1
	%	0			4			6			7			5		

T: Treated, NT: Non treated.

**Table 4 brainsci-13-00753-t004:** Volumes of the corpus callosum and the lateral ventricles.

Group	Corpus Callosum		RLV		LLV	
(Gestation Weeks)	Mean	S.D.	Mean	S.D.	Mean	S.D.
25–27	2.211	2.355	7.912	6.740	6.960	4.599
28–31	1.162	1.465	8.880	12.422	6.460	7.966
32–34	2.448	2.699	7.288	6.364	6.199	5.281
35–37	2.058	2.146	8.409	10.922	5.849	4.380
25–37	1.876	2.204	8.189	10.016	6.251	6.056

**Table 5 brainsci-13-00753-t005:** Bayley’s scores at 3 years of age.

Bayley’s Scores	Treated (Mean)	Non-Treated (Mean)	Min—Max	Permutation *p* Value
MDI	100	79	50–134	0.001
PDI	104	81	50–127	0.001

**Table 6 brainsci-13-00753-t006:** Regression Analyses, significant predictors for MDI and PDI.

Predictors	MDI			PDI		
	Coefficient	Standard Error	*p*	Coefficient	Standard Error	*p*
Infant Sepsis	−4.49	1.750	0.010	−5.419	1.779	0.002
Apgar < 7				−4.202	1.757	0.017
Corpus Callosum Volume	3.359	1.705	0.050	5.474	1.792	0.002
Left Lateral Ventricle Volume	−7.702	1.693	<0.001	−4.955	2.000	0.014
Right Lateral Ventricle Volume				−5.750	1.794	0.001

## Data Availability

Deidentified data will be provided upon written request to the authors who originated the data.

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
