# Peer review of "Prevention of Neurological Sequelae in Preterm Infants"

_brainsci, 2023, doi:10.3390/brainsci13050753_

Round 1

Reviewer 1 Report

This paper is interesting, overall well written and well documented. This study explored the prenatal and perinatal risk factors of brain damage in preterm infants. Moreover, this paper confirmed the beneficial effect of the early Katona´s neurohabilitation procedure to prevent neurologic and cognitive sequels in preterm infants with prenatal and perinatal risk factors for brain damage.

The goal (Line 91-93) of this research and conclusions (Line 34-36, Line 405-411) did not correspond well.

Author Response

Dear Reviewer #1, we would like to thank you for taking the time to review our manuscript. Your comments and suggestions were insightful and greatly helped us improve the manuscript. In response to your feedback, we have made changes to the manuscript. Below is a summary of our responses:

The original manuscript stated the following:

  • Lines 91-93:The main goal of this study is to explore the predictive value of risk factors and quantitative analyses of MRI and visual and auditory evoked responses for Bayley´s outcome.
  • Lines 34-36: Conclusions: Katona’s neurohabilitation treatment significantly improves neurodevelopment in preterm infants, probably via the prevention of neurologic and cognitive sequels produced by brain risk factors.
  • Lines 405-411:Conclusions: We confirmed the beneficial effect of the early Katona´s neurohabilitation procedure to prevent neurologic and cognitive sequels in a new sample of preterm infants with prenatal and perinatal risk factors for brain damage. MRI studiesalsoconfirmed that infants with prenatal and perinatal risk factors for brain damage present structural abnormal findings. These data support the suggestion that early Katona’s neurohabilitation procedure should be used in preterm infants with prenatal and perinatal risk factors for brain damage to prevent neurologic and cognitive sequalae.

We modify the goals and the conclusions based on your valuable feedback:

  • As reported in other studies, our main objective was to investigate the Bayley’s outcome at 3 years old in a sample of 166 preterm infants, 128 of whom received Katona’s therapy, and 38 received no treatment. All children had prenatal and perinatal risk factors for brain damage.
  • The main goal of this study is to explore the predictive value of risk factors and quantitative analyses of MRI obtained before 4 months old for Bayley´s outcome at 3 years old.
  • Conclusions:

1) The results indicate that preterm infants who received Katona’s neurohabilitation procedure exhibited significantly better outcomes at 3 years old compared to those who did not receive the treatment.

2) The presence of sepsis and the volumes of the corpus callosum and lateral ventricles at 3-4 months were significant predictors of the outcome at 3 years old.

Thank you very much for your helpful feedback, which assisted us in improving our manuscript.

Best regards,

Thalía Harmony

Reviewer 2 Report

The authors present a study in favor of the neurohabilitation of Katona. Even if we could be convinced of the relevance of this procedure, the overall scientific quality appears limited.

The conclusion included in the abstract (but also at the end of the paper) "Katona’s neurohabilitation treatment significantly improves neurodevelopment in preterm infants, probably via the prevention of neurologic and cognitive sequels produced by brain risk factors" could appear as speculation for at least two reasons:

1-As brain continue to maturate during all types of learning during infancy (and after three-years of age) the time point of comparison (3 years) should be clearly integrated in this conclusion.

2-Since authors could not control all environmental factors of each infant during the period, the presented link between the improvement and the "prevention of neurologic and cognitive sequels produced by brain risk factors" should be softened.

Authors assume themselves these points in lines 52-53 "The factors that influence neurodevelopment in infants born preterm are multiple, contributing to the complexity of follow-up research and variations in reported outcomes [6]".

I don't think the methodological presentation lines 70-72 should be listed in the introduction, but rather in the method section. In a comparable aspect, lines 74-78 would be better in the discussion.

In a general point of view, the second half of the introduction gives a list of technical choices for investigations. Don't think that this is relevant for the overall scientific quality of the paper. I suggest to present the scientific knowledge of the literature, rather than a list of techniques. Should be completely rewritten.

In the section 2.2., the design is presented. But (1) what was considered as a "predictive design"? Should explain better, and (2) did authors considered a blind or double-blind method to treat data ?

The risk factors presented in tables are not defined in the method section. A short description of what was clinically considered for each factor should be listed.

In tables 4, 5 and 6, authors chose to present number of infants in each of their chosen categories, inside each type of investigation. It is very surprising. Why authors did not choose to present the real measurements obtained ? For example volumes for MRI, values for evoked responses...

Why basic clinical data of growing infants are not available? Body weights, blood samples with biochemical markers for both mothers and infants (folate, B-vitamines are known to strongly influence brain development and outcomes). PO2 would be also very helpful. Surprisingly, authors discuss of that in the beginning of the discussion, but without any of these types of markers. If these markers are not available, all the beginning of the discussion (until line 350) should be soften.

If authors could not improve the scientific quality adding some measurements rather than only number of infants in chosen groups of measures, it should be explain in the 4.4. section "limitation of the study".

Author Response

Answers to Reviewer 2

Dear Reviewer #2, We would like to express our sincere gratitude for dedicating your time to review our manuscript. Your invaluable comments and suggestions provide insightful feedback, which helped us significantly enhance the quality of our manuscript.

In response to your feedback, we have made several changes to the manuscript. Below is a summary of your comments and our responses:

Reviewer 2:

The conclusion included in the abstract (but also at the end of the paper) "Katona’s neurohabilitation treatment significantly improves neurodevelopment in preterm infants, probably via the prevention of neurologic and cognitive sequels produced by brain risk factors" could appear as speculation for at least two reasons:

1-As brain continue to maturate during all types of learning during infancy (and after three-years of age) the time point of comparison (3 years) should be clearly integrated in this conclusion.

2-Since authors could not control all environmental factors of each infant during the period, the presented link between the improvement and the "prevention of neurologic and cognitive sequels produced by brain risk factors" should be softened.

Authors assume themselves these points in lines 52-53 "The factors that influence neurodevelopment in infants born preterm are multiple, contributing to the complexity of follow-up research and variations in reported outcomes [6]".

Answer: Based on your suggestions, we have modified the conclusion as follows:

Conclusions:

1) The results indicate that preterm infants who received Katona’s neurohabilitation procedure exhibited significantly better outcomes at 3 years old compared to those who did not receive the treatment.

2) The presence of sepsis and the volumes of the corpus callosum and lateral ventricles at 3-4 months were significant predictors of the outcome at 3 years old.

(lines 33-36 and 387-391).

Reviewer 2:

I don't think the methodological presentation lines 70-72 should be listed in the introduction, but rather in the method section. In a comparable aspect, lines 74-78 would be better in the discussion.

In a general point of view, the second half of the introduction gives a list of technical choices for investigations. Don't think that this is relevant for the overall scientific quality of the paper. I suggest to present the scientific knowledge of the literature, rather than a list of techniques. Should be completely rewritten.

Answer: In response to your comments, we have removed the paragraphs you mentioned from the introduction and incorporated relevant information into the Methods section.

Reviewer 2:

In the section 2.2., the design is presented. But (1) what was considered as a "predictive design"? Should explain better, and (2) did authors considered a blind or double-blind method to treat data?

Answer: In response to your comments, we have removed the paragraph regarding the design of the study. We would like to clarify that the results presented in the manuscript, with the exception of the Bayley's indices, were obtained prior to treatment. To ensure the objectivity of the Bayley’s test, the psychologist conducting the assessments was not aware of which infants had received treatment.

Reviewer 2: The risk factors presented in tables are not defined in the method section. A short description of what was clinically considered for each factor should be listed.

Answer: As part of the clinical evaluation, we employed expert neuropediatricians who clinically studied all infants following the recommendations of Amiel-Tison [18]. These specialists had access to the medical record of each infant, including data from the pregnancy, labor, and the newborn period which included prenatal and perinatal risk factors observed in the hospital. (lines 114-121)

Furthermore, to provide additional clarity and context, we have included a paragraph about the relevance of the risk factors before each table for easy reference (lines 193-194; lines 198-203).

Reviewer 2: In tables 4, 5 and 6, authors chose to present number of infants in each of their chosen categories, inside each type of investigation. It is very surprising. Why authors did not choose to present the real measurements obtained ? For example volumes for MRI, values for evoked responses...

Answer: We have added a new table (Table 4, lines 231-232) which shows the volumes of the corpus callosum and the lateral ventricles. In a new paragraph, we have also described the MRI abnormalities data (lines 217-229).

We have removed all data related to evoked potentials from the manuscript. This decision was based on the fact that no all the infants had the evoke potentials performed due to the difficulty in obtaining reliable measurements due to infant movements, and visual responses which require the infant to have their eyes open and be attending to the stimuli. Additionally, brainstem auditory evoked responses were obtained during sleep, resulting in a limited number of recordings without artifacts. Furthermore, the results of the regression analysis in the previous version of the manuscript showed that evoked potentials were not significant predictors of Bayley’s indices. Therefore, we decide to eliminate these data from the reviewed manuscript. 3 references to evoked potentials were also removed.

Reviewer 2: Why basic clinical data of growing infants are not available? Body weights, blood samples with biochemical markers for both mothers and infants (folate, B-vitamins are known to strongly influence brain development and outcomes). PO2 would be also very helpful. Surprisingly, authors discuss of that in the beginning of the discussion, but without any of these types of markers. If these markers are not available, all the beginning of the discussion (until line 350) should be soften.

Answer: We have added the following paragraph:

"At 3 years old, clinical evaluations were conducted on preterm infants across four groups. Among the extremely preterm group, 8 of 9 treated infants had typical cognitive and motor development, while only 1 out of 4 untreated infants showed adequate development. Similarly, 32 out of 42 treated infants and 11 out of 15 untreated infants in the very preterm group had typical cognitive and motor evaluations. In the moderate preterm group, the percentage of treated (33 out of 41 ) and untreated (11 out of 15) infants with typical cognitive evaluations was similar. However, untreated infants exhibited motor impairments such as coordination deficits and monoparesis. In the late preterm group, untreated infants showed more frequent deficits in both cognitive and motor areas than treated infants." (lines 268-276)

We also softened the beginning of the discussion, as you suggested.(lines 296-319)

Thank you very much for your helpful feedback, which assisted us in improving our manuscript.

Best regards,

Thalía Harmony

Round 2

Reviewer 2 Report

In my opinion, you have improved your manuscript